# Recombinant Human Growth Hormone Therapy for Childhood Trichorhinophalangeal Syndrome Type I: A Case Report

**DOI:** 10.3390/children9101447

**Published:** 2022-09-22

**Authors:** Dan Huang, Jia Zhao, Fang-Ling Xia, Chao-Chun Zou

**Affiliations:** 1Department of Child Health Care, Hangzhou Women’s Hospital (Hangzhou Maternity and Child Care Hospital), Hangzhou 310008,China; 2Department of Endocrinology, The Children’s Hospital, Zhejiang University School of Medicine, Hangzhou 310052, China; 3Department of Pediatrics, Zhuji People’ Hospital of Zhejiang Province, Shaoxing 311800, China

**Keywords:** trichorhinophalangeal syndrome type I, *TRPS1* gene, growth retardation, rhGH therapy

## Abstract

Trichorhinophalangeal syndrome type I (TRPS I; MIM 190350) is a rare autosomal dominant disorder of congenital malformations due to variants of the gene *TRPS1*. We reported on an 11-year-old Chinese boy with TRPS I. He had typical clinical findings, including sparse hair, a bulbous nose, a long philtrum, a thin upper lip, and skeletal abnormalities including cone-shaped epiphyses, shortening of the phalanges, and short stature. Trio whole exome sequencing identified a likely pathogenic heterozygous variant c.1957C > T (p.Q653*) in exon 4 of *TRPS1*, which has not been previously reported. He had been treated with rhGH therapy at a dose of 0.34 mg/(kg/week) at age 11, and a follow-up was conducted for one year. The rhGH therapy led to an increase in growth with a mean growth velocity of 1.12 cm/month (+1.1 SDS/year), and insulin-like growth factor 1 (IGF-1) concentration increased within normal range in our case. Moreover, we summarize 12 cases with TRPS I, including *TRPS1* gene variants, growth hormone (GH) axis evaluation, IGF-1 concentration, and treatment in each analyzed case. Eight cases with TRPS I show a good response to rhGH therapy, and five of them have elevated IGF-1. Classic GH deficiency is not common among patients with TRPS I. The presence or absence of GH deficiency is not an absolute criterion for determining whether rhGH therapy should be used in TRPS I. It proves that rhGH therapy improves height outcomes before puberty in TRPS I in the short term. Effects on final adult height will need a longer follow-up and more adult-height data. The rise in IGF-1 could correlate with an increase in short-term height. Measuring IGF-1 levels is recommended as part of the assessment during the follow-up of patients with TRPS I.

## 1. Introduction

Trichorhinophalangeal syndrome (TRPS) is an autosomal dominant congenital disease with high penetrance and extensive phenotypic variability [1]. It is a rare syndrome that was first reported in 1966 with a prevalence of 0.2 to 1 per 100,000 [2]. Many patients with TRPS may be undiagnosed, so unbiased population-based estimates of the prevalence of TRPS are not available. Based on clinical manifestations and genetic changes, it can be divided into three types, including TRPS type I (TRPS I; MIM 190350); TRPS type III (TRPS III; MIM 190351), which is due to *TRPS1* gene variants; and TRPS2 (TRPS II; MIM 150230), which is a syndrome involving deletions of both *TRPS1* and EXT1 genes [3]. TRPS I is a condition characterized by sparse hair and craniofacial and skeletal abnormalities. The clinical features are heterogeneous, sparse, and slow-growing scalp hair; sparse lateral eyebrows; a bulbous nose; a thin upper lip; brachydactyly; protruding ears; and dental abnormalities that have been reported [4]. In addition, cone-shaped epiphyses, noteworthy at the phalanges, are the most typical radiographic findings in TRPS I, which are not predominantly detectable before the age of 2 [5]. Both TRPS II and III have type I features, but TRPS II is often associated with multiple exogenous osteochondroma and intellectual disability. TRPS III is often associated with severe short fingers (toes), short stature, and severe growth retardation [6]. Overall, type I is the mildest, but type III is the most severe.

TRPS I often corresponds to distinct variants in the *TRPS1* gene, which is located on human chromosome 8q23.3. The *TRPS1* gene consists of a total of seven exons and encodes a 1294-amino acid transcription factor that represses GATA-regulated genes and binds to a dynein light chain protein [7,8]. The binding of the encoded protein to the dynein light chain protein affects the binding to GATA consensus sequences and suppresses transcriptional activity [9,10]. These mutants likely have a decreased affinity for DNA, because of the altered GATA zinc fingers because they and exert a dominant negative effect [11]. The *TRPS1* gene is a zinc finger transcriptional repressor involved in the regulation of the development of chondrocytes, bone perichondrium, and hair follicles [12,13]. To date, more than 140 pathogenic variants in TRPS have been identified, and these are distributed along the entire coding region of the gene, among which, exon 4, 5, and 7 variants are more common. It was also found that patients with missense variants tend to be more severely affected. Due to the lack of understanding of the disease, it is often misdiagnosed or there are missed diagnoses. The rhGH therapy in children with TRPSI has rarely been reported. Moreover, previous reports that have shown that the early initiation of rhGH therapy is associated with better height outcomes remain controversial. Herein, we report an 11-year-old Chinese boy with TRPS I with the typical clinical manifestations and a likely novel pathogenic nonsense variant in exon 4 of *TPRS1*, who had an excellent growth response to rhGH therapy in the short term. Then, we reviewed the related literature to summarize genetic characteristics and the efficacy of rhGH in all related TRPS I cases, which may guide further rhGH therapy in patients with TRPS I in the future.

## 2. Case Report

### 2.1. Case Presentation

An 11-year-old boy was referred to the endocrinology service for the evaluation of his short stature and extremely slow growth velocity. He was the first child of non-consanguineous Chinese parents. The child was born prematurely at 36 weeks of gestation by cesarean with a birth weight of 2.4 kg, −1.15 standard deviation scores (SDS), and a birth length of 48 cm, −1.3 SDS. The birth head circumference parameters were not available, but they were described as normal. No milestones in motor skills and language retardation were reported. However, the low growth velocity and the retardation of bone age were noted after birth. His height was 93.3 cm (−4.0 SDS) at the age of 4.9 years old, according to the height growth curves for children and adolescents in China [14,15]. We used two different methods: arginine and clonidine, for the GH stimulation test, and the peak was 18.17 ng/mL and 15.10 ng/mL, respectively, with normal IGF-1 at the age of 5 years old. Although the Wechsler Intelligence Test was performed at the age of 10 years old and was normal with an IQ of 112, his parents noted that he is better at math and has an average grade in Chinese. The family history is negative for short stature. The height of his father was 172 cm, and the height of his father mother was 160 cm. The family history revealed that his parents and younger sister were all healthy.

### 2.2. Physical Examination

Physical examinations show a height of 132.0 cm (−2.0 SDS) and a weight of 30.1 kg. His BMI was 17.2 kg/m^2^, and he was at Tanner Stage II (G2, PH1) for pubic hair and testicle volume at the age of 11. Prominent craniofacial features and ectodermal dysplasia are noted, including relatively sparse scalp hair, a high forehead hairline, laterally thinned eyebrows, broad nasal cartilage, a bulbous tip of the nose, a long and flat philtrum, a thin upper lip vermilion, a high palate, retromicrognathia, downslanting palpebral fissures, and large laterally protruding ears (Figure 1A). Oral examination presents delayed dentition and irregular tooth alignment in his permanent teeth (Figure 1B). The examination of both hands shows very mild, short, and wide middle fingers, and the second to fourth fingers of both hands are mildly tumid (Figure 1C,D). Features of his feet include a short and thick big toe deformity, ten wide, short, and thin toenails, and being flat-footed (Figure 1E). His skin presents xanthomas on his knee and ankle joints (Figure 1F). In the bone, muscle, or joint, there is physiological bending of the backbone and no tenderness or vertically percussed pain. The rest of the systemic examination, including the cardiac, respiratory, abdominal, and nervous systems, is normal.

### 2.3. Diagnostic Assessment

The thyroid function, blood cell count, serum calcium, 25-OH vitamin D3, liver function, kidney function, trace element examination, metabolic examination determination of glucose, and insulin function were all normal. Sex hormone levels, including follicle-stimulating hormone levels, luteinizing hormone levels, and testosterone levels, were, respectively, 4.7, 1.45, and 1.47 IU/L. The plasmatic concentration of IGF-1 and IGF binding protein 3 (IGFBP-3) were normal in relation to age and sex (365 ng/mL and 2.0 μg/mL, respectively).

The radiograph of the left hands showed pollex phalanx epiphyses; particularly, the first proximal phalanges are characterized by cone-shaped epiphyses (circle) and are embedded with thickening of surrounding soft tissue; shortness of middle phalanges; and the middle phalanges of the third to fourth finger had a fusion of diaphysis and epiphysis. It also showed a bone age of about 9 years old according to the Greulich and Pyle method (Figure 2A). Furthermore, the anteroposterior and lateral views of the foot showed that the first, third, and fifth proximal phalanges of both feet were conical epiphyses (Figure 2C–F). The pelvis showed no obvious abnormal X-ray signs (Figure 2G). Testicular volume by ultrasound on the left was 3.54 mL and on the right was 3.24 mL. An MRI showed the normal size and location of the pituitary gland and stalk. No pathological abnormalities were found by heart, thyroid, and abdominal ultrasound.

Trio whole exome sequencing (Trio-WES) for the proband and his parents were performed in our unit. SangerTrio-WES revealed a heterozygous c.1957C > T variant in exon 4 of the *TRPS1* gene in our case, which was not found in his parents and implied a de novo. The variant was confirmed by Sanger sequencing (Figure 3). It is a nonsense variant and results in an immediate stop codon (p.Q653*), leading to the premature termination of the polypeptide chain synthesis and the premature truncation of the TRPS1 protein.

The classification of this sequence variant according to the ACMG guidelines: a. It has not been reported in normal population controls (ESP database, 1000-person database, ExAC database), and the undetected variants belong to the evidence of moderate pathogenicity (PM2). b. It is a nonsense variant, which leads to the premature termination of protein translation, and the expression of the termination protein still affects its function, which belongs to the evidence of moderate pathogenicity (PM4). c. It is not detected in the sequencing data of the parents, which was analyzed as a new variant of the child and belonged to strong pathogenic evidence (PS2). The pathogenicity rating combined with the above variant evidence and evidence classification, according to the ACMG guidelines, classified that this variant has two moderate-pathogenicity evidence and one strong-pathogenicity evidence: 1 PS + 2 PM, which means it is a likely pathogenic variant.

### 2.4. rhGH Intervention and Follow-Up

Although the GH peak was normal in the GH drug stimulation, rhGH therapy was initiated after the genetic diagnosis was confirmed. The initiated dose was approximately 0.34 mg/(kg/week). The IGF-1 was 365 ng/mL (ref. 169–562 ng/mL) before rhGH therapy and then ranged from 500 to 553 ng/mL. The rhGH therapy led to a significant increase in growth velocity with a mean growth velocity of 1.12 cm/month (+1.1 SDS/year) during the 12 months of rhGH therapy (Figure 4). The progress of his bone age was not obvious after 6 months of rhGH therapy (Figure 2B). No adverse effects such as scoliosis and hyperglycemia were found at the 3 to 6 month intervals.

### 2.5. Literature Review

We reviewed the literature that used “trichorhinophalangeal syndrome type I” or “TRPS I” and “growth hormone” in English (PubMed database and Web of Science) and Chinese (CNKI and Wan Fang database). Besides our case, there were 11 additional TRPSI cases with rhGH therapy that were reported. After reviewing the reported cases, we found that eight cases (cases 1–4 and 6–9) provided a peak of definite GH that was confirmed by two different GH stimulation tests. Among these twelve cases, seven had GH deficiency (GHD), including two who had partial GHD (cases 6 and 7), five had GHD (cases 8–12), and five (cases 1–5) had normal GH. Moreover, eight cases (cases 1–4 and 6–9) showed a good response to rhGH therapy, including the two with partial GHD and one with complete GHD (case 8). Case nine shows an adult with complete GHD with a slight effect. Five of them (cases 1–3, 8, and 9) had elevated IGF-1. Meanwhile, the other four cases (cases 5 and 10–12) showed no improvement in growth velocity, as shown in Table 1.

## 3. Discussion

The diagnosis of TRPS is mainly based on clinical and radiographic features. The genotype–phenotype correlations in TRPS I have been a controversial issue. It is observed that prominent variabilities exist within and among families with the same TRPS I pathogenic variants [16]. TRPS may still remain challenging and undiagnosed for many [17]. Here, we report a case of this disease in an 11-year-old Chinese boy with typical features, which conform to the features of TRPS I. Molecular testing confirmed the diagnosis of TRPS I in this boy with a novel heterozygous nonsense variant. According to the ACMG guidelines, this variant is a likely a pathogenic variant. It was not possible to perform functional studies to analyze the TRPS1 protein expression.

Growth in children with TRPS is characterized by decreased growth velocity and high-penetrance skeletal abnormalities. Growth retardation is common and considered a progressive process in TRPS I [18]. Data demonstrate that TRPS has a broad spectrum with regard to brachydactyly and growth retardation caused by the premature closure of bone growth plates [3]. The abnormal, premature fusion and maturation of the epiphysis can also lead to growth retardation. TRPS I is mainly caused by *TRPS1* gene variants. Piranit Nik et al. found that *TRPS1* regulates the expression of a variety of Wnt inhibitors and transcription factors during vibrissa follicle morphogenesis in mice [18]. *TRPS1* regulates the structure of the growth plate by inhibiting Stat3 and Pthlh signal transduction and promoting the differentiation and proliferation of columnar chondrocytes [19]. A possible explanation could be that TRPS1 is associated with the longitudinal growth of all long bones and abnormal growth plate morphology by regulating the number of cells in the proliferative zone and the hypertrophic zone. In addition, osteopenia may be present in TRPS I. *TRPS1* has been shown to modulate mineralized bone matrix formation in different osteoblast cells as a regulator of osteocalcin transcription [20]. Recent data have indicated a role for TRPS I in osteoblast differentiation. Some studies suggest the possible role of *TRPS1* in bone homeostasis, especially in determining bone mineral density, but that needs further study.

Through the literature review, we found 11 cases of TRPS with rhGH therapy reported previously (Table 1) [21,22,23,24,25,26,27]. In these cases, patients with similar clinicopathologic characteristics benefit variably from rhGH therapy. It is clear that there is variable GH sensitivity among TRPS children with short stature. It was shown that rhGH therapy was effective in improving height velocity in eight cases (66.7%) with TRPS I, and the height SDS increase was 0.4–1.95. GHD was diagnosed in seven of these cases, among which, two were considered as partially GH deficient. The therapy in two complete GHD cases (cases 8 and 9) and two cases (cases 6 and 7) with partial GHD turned out to be effective. Factors contributing to case 9 with a slight response to rhGH threapy may include the low rhGH dose (0.2 mg/kg/week) and the adult with complete GHD. Among the four cases, there was less increased height; the reasons are complicated. Some TRPS symptoms may mimic growth hormone deficiency and other endocrine disturbances. Patients with TRPS I may have GHD since this transcription factor has been found to be expressed in the pituitary and hypothalamus [28]. This suggests that short stature in TRPS I could partly be due to the presence of GHD. Additionally, therapeutic effects are particularly associated with internal GH status. One reason for case 5 is most likely that treatment was initiated at a bone age of 16 years, a mature bone age. The other two reported cases of TRPS I with GHD did not respond to growth hormone therapy. Cases 10 and 11 (a pair of twin girls with TRPS I) presented with poor growth but lacked the actual GH stimulation test results and Tanner stage. It is confusing why the efficacy of rhGH is not good in TRPS I cases (cases 10 and 11) with GHD. This may be related to growth hormone resistance. Similar conditions were reported after GH treatment in children with GH-sufficient, idiopathic short stature [29]. As a result, it is more difficult to regulate the status of GH secretion in patients with genetic syndromes than in the normal population. The GH stimulation tests for the diagnosis of GHD lack specificity, especially in pre-pubertal children who have not been primed. The classic GH deficiency is not common among patients with TRPS. Defining GHD is a complex process requiring comprehensive clinical assessment combined with biochemical tests and radiological evaluation. Multiple factors affect the growth response to GH, including dose dependence, age at initial treatment, GH sensitivity, parental height, and delayed bone age, many of which are unknown. The growth response to GH therapy could not be a clinical marker for GHD, as shown in our patient. Furthermore, the presence or absence of GH deficiency is not an absolute criterion for determining whether rhGH therapy should be used for TRPS I.

**Table 1 children-09-01447-t001:** A summary of 12 cases with *TRPS1* gene variants, GH axis evaluation, and treatment in each analyzed patient with TRPS I.

No	Age/Gender	Variants	GH Status	BA	IGF-1	GH Dose (mg/kg/week)	IGF-1 (ng/mL)		Height SDS		Growth Velocity	Adult Height	Ref
							On	End	On	End			
1	11 Y/Male	c.1957C > T (p.Q653*)	Normal ^b^	9 Y	Normal	0.34	365	500–533↑	−2.0	−0.9	1.12 cm/month +1.1 SDS/1 Y	ND	Current
2	7 Y/Male	NA	Normal ^c1^	4 Y	low	0.3–0.43	81	467↑	−3.18	−1.37	+1.81 SDS/over 3 Y	NA	[22]
3	7 Y/Female	NA	Normal ^c2^	6 Y 5 M	low	0.35–0.54	81	347↑	−3.05	−1.1	+1.95 SDS/over 2 Y	NA	[22]
4	10 Y/Male ^a^	c.3368 G > A (p.Trp1123Stop)	Normal ^a^	6.4 Y	Normal	0.27–0.29	180	ND	−2.4	−1.4	7.7 cm/Y +1 SDS/ over 2 Y	NA	[27]
5	14 Y/Male	c1630C > T (p.Arg544X)	Normal	16 Y	Normal	NA	591	ND	−2.1	−3.6	<1 cm over 6 months	−3.6	[21]
6	12 Y Male	NA	Partial ^d1^ GHD	9 Y 6 M	NA	0.26	NA	NA	−2.2	−1.5	+0.7 SDS/over 5 Y	−1.5	[24]
7	9 Y 9 M/Male	c2722C > T (p.R908X)	Partial ^d2^ GHD	7 Y 8 M	low	0.26	86.9	ND	−2.0	−0.1	+1.9 SDS/over 7 Y	−0.1 Near adult final height	[24]
8	3.5 Y/Male	c.3698G > A(p.Cys1233Tyr)	GHD ^e^	2.8 Y ^e^	low	0.21	30.1	452↑	−3.2	−1.7	+1.5 SDS/over 5 Y	NA	[26]
9	17 Y/Female	c.2520dupT (p.Arg841LysfsX3)	GHD ^f^	2.6 Y ^f^	Normal	0.21	78.5	561.2↑	−2.5	−2.1	5.7 ± 0.9 cm/Y 0.4 SDS/over 10 Y	−2.1	[21]
10	8.3 Y/Female Twins, first born	NA	GHD	5.7 Y	low	0.21	ND	ND	−2.6	−3.1	No increase	−3.1	[23]
11	8.3 Y/Female Twins, second born	NA	GHD	5.7 Y	low	0.21	ND	ND	−2.4	−3.4	No increase	−3.4	[23]
12	10 Y/Female	c.1198C > T (p.Gln400X) c.2086C > T (p.Arg696X)	GHD	4.5 Y	low	ND	ND	ND	−2.0	NA	+1.5 cm/Y	NA	[25]

Y, years; M, month; GHD = growth hormone deficiency; BA = bone age; ND, not determined; NA, not reported; ↑, increased; ^a^, a family with TRPS I and one child with rhGH therapy; GH stimulation tests with arginine and insulin (peak GH of 15.3 ng/mL and 8.2 ng/mL, respectively); ^b^, GH stimulation tests with arginine and clonidine (peak GH of 18.17 ng/mL and 15.10 ng/mL, respectively); ^c1^, GH stimulation tests with arginine and clonidine (peak GH of 15.6 ng/mL and 4.5 ng/mL, respectively); ^c2^, arginine and clonidine (peak GH of 22.5 ng/mL and 20.0 ng/mL, respectively); ^d1^, GH stimulation tests with arginine and insulin (peak GH of 6.8 ng/mL and 12.7 ng/mL, respectively); ^d2^, GH stimulation tests with arginine and insulin (peak GH of 10.2 ng/mL and 5.4 ng/mL, respectively); ^e^, GH stimulation tests with clonidine and glucagon (peak GH of 3.3 mcg/L and1.7 mcg/L, respectively); chronologic age 4.8 Y; ^f^, two tests: ① GH stimulation tests using insulin and levo-dopamine (peak GH of 3.17 ng/mL and 5.08 ng/mL, respectively); ② adult GH deficiency: GH stimulation tests using glucagon and levo-dopamine (peak GH of 4.65 ng/mL and 4.84 ng/mL, respectively); chronologic age 4 Y.

The outcomes of follow-up results for TRPS I remain dismal despite improvements in treatment and management strategies for short stature. Ludecke et al. reported the average height of 75 patients with TRPS to be 1.41 SD below the average height of the respective population (SD = 1.15) [30]. During the period of 5–9 years old, his height was under −2 SDS for a long time. It is known that the skeletal characteristics of TRPS are delayed bone age before puberty, accelerated bone age after puberty, and adult-onset chronic joint pain. Untreated patients with TRPS achieve a poor average adult height. Therefore, the best initiation time for patients for whom rhGH therapy is considered should be before puberty to achieve a good result. Several cases suggested that rhGH therapy could improve growth in TRPS children with growth retardation [22,24,26,27]. Our case proves that GH improves height outcomes in TRPS I in the short term due to the lack of final height data. Reviewing the reported cases, there are only six cases (cases 5–7 and 9–11) that reported final height data. In three cases (cases 5, 10, and 11), the outcomes of the final height are unfavorable. Case 9 shows an adult with complete GHD with a slight effect, and there is a family history of short stature [21]. Four cases (cases 1–4) with good short-term responses had non-GHD and no adult height. Thus, more data are needed before TRPS I can be considered an indication for GH treatment. There is insufficient proof that the acceleration of the growth rate by rhGH in non-GH-deficient children will result in increased adult height. Analysis of the long-term effect of rhGH therapy on the increase in final adult height in children with TRPS and possible late-onset complications is expected in the future. The mechanism of the GH effect in TRPS has yet to be elucidated, and several hypotheses have been proposed. One effective mechanism is that GH therapy may increase the concentration of IGF-1 throughout the body, thus enhancing the effectiveness of IGF-1 in growth plate action. Indeed, we found that the IGF-1 concentrations of five cases, including our case, had increased during rhGH therapy within the low or normal range. The receptors for IGF-1 are expressed on chondrocytes in the epiphyseal growth plate [31]. So, IGF-1 may play a supplementary role in TRPS with GH therapy and may have a beneficial influence on the epiphyses of children with TRPS. On the other hand, the GH receptor and the GH binding protein are expressed in the growth plate in early maturing chondrocytes between the proliferative and hypertrophic zones [21]. This indicates that GH may have a direct effect on the stimulation of longitudinal bone growth. We cannot rule out the possibility that growth hormone deficiency is cardinally related to TRPS. Although the true nature of the GH-IGF-1 axis defect is not clearly understood in these patients, partial GH resistance may be a possible mechanism [22]. The initiation of rhGH therapy may benefit both height and bone mineral density with improved epiphysis status in children with growth retardation [22]. The changes in IGF-1 in TRPS cases before and after rhGH treatment were summarized. Five of them (cases 1–3, 8, and 9) had elevated IGF-1. Moreover, the GH peak value was normal in cases 2 and 3, but their IGF-1 level was low. Our case did not have GH deficiency with IGF-1 concentration increasing within the normal range. They all have an excellent effect on rhGH therapy. Investigations of rhGH therapy effect and changes in serum IGF-1 level in these kinds of patients are rare, and treatment-related IGF-1 data have not been validated in long-term studies, but the rise in IGF-1 correlates with short-term height increase. Measuring IGF-1 levels is recommended as part of the assessment during the follow-up of patients with TRPS I. Moreover, a larger scale of the study is warranted to understand the contribution of the GH-IGF-1 axis to short stature in TRPS I.

## 4. Conclusions

In summary, in children with sparse scalp hair, a high hairline, a pear-shaped nose, and short stature, TRPS I should be considered in the differential diagnosis, and genetic detection should be performed early. A new likely pathogenic heterozygous variant c.1957C > T (p.Q653*) was found in an 11-year-old child. rhGH therapy before puberty could be effective in improving short stature in patients with TRPS I in the short term. The rise in IGF-1 could correlate with a short-term height increase. Measuring IGF-1 levels is recommended as part of the assessment during the follow-up of patients with TRPS I.

## Figures and Tables

**Figure 1 children-09-01447-f001:**
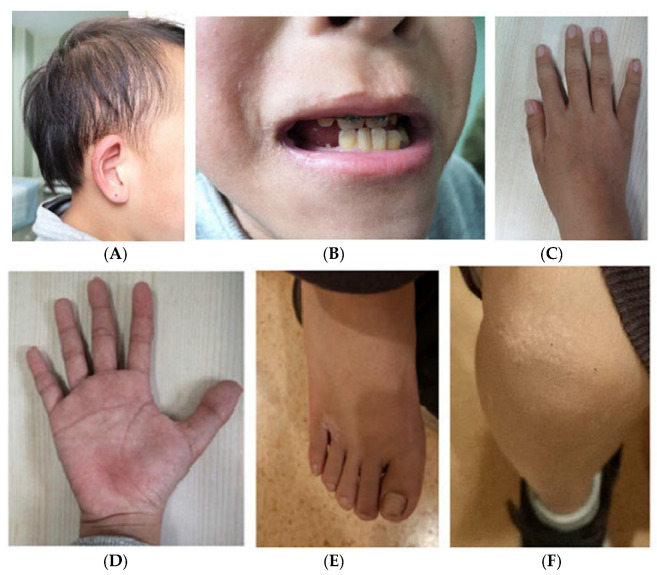
Clinical manifestations of the child: (**A**): Specific facial features, including thin scalp hair, a high hairline, and large ears. (**B**): Thin upper lip and small lower jaw. The teeth are not arranged in order and the bite is not correct. (**C**,**D**): Hands of the boy: The 2nd to 4th fingers of both hands are mildly tumid, the middle fingers of both metacarpals in the hand are short and wide, and short thumbs; (**E**): feet are characterized by a short and thick big toe deformity, ten wide, short, and thin toenails; (**F**): skin xanthomas on his knee joint.

**Figure 2 children-09-01447-f002:**
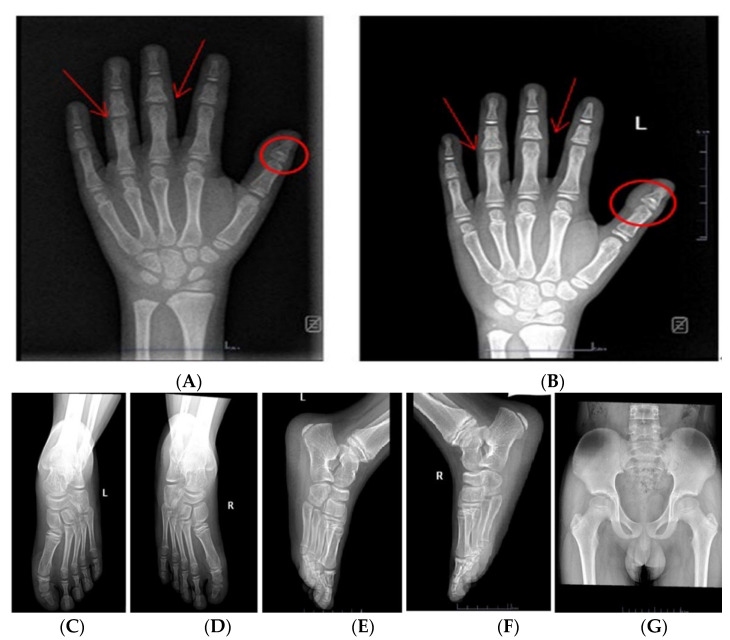
X-ray report of the child: Frontal radiographic findings of the left hands of the child: (**A**): The photograph of the hands shows brachydactylia with shortening of the middle phalanges of all fingers. The coned epiphyses of the first proximal phalanges (circles) and more subtle, partially fused coned epiphyses of the third to fourth middle phalanges (arrows) can be noticed. Bone age (9 years) was delayed compared to chronological age (11 years). (**B**): The radiograph of the left hand notes a chronological age of 11 years and 6 months. (**C**,**D**): Anteroposterior radiograph of the foot; (**E**,**F**): Lateral radiograph of the foot: Foot X-ray for him demonstrates that the 1st, 3rd, and 5th proximal phalanges of both feet were conical epiphyses. (**G**): Plain radiographs of the anteroposterior view of the pelvis: no obvious abnormal X-ray signs in the pelvis.

**Figure 3 children-09-01447-f003:**
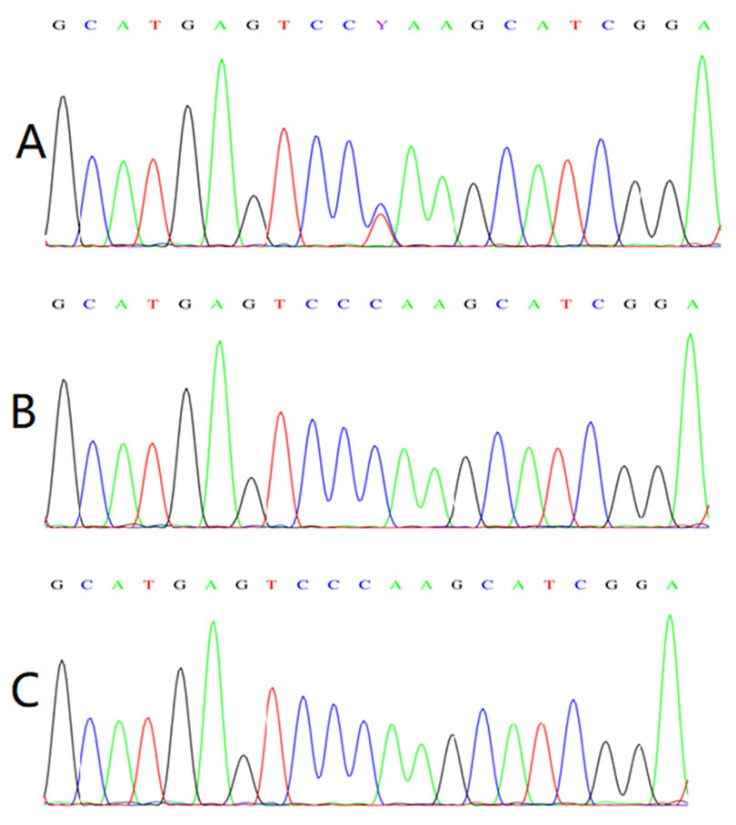
(**A**): He had a novel heterozygous nonsense variant of c.1957C > T (p.Q653*). (**B**,**C**): Variant analysis of his father and mother showed no variants in the *TRPS**1* gene.

**Figure 4 children-09-01447-f004:**
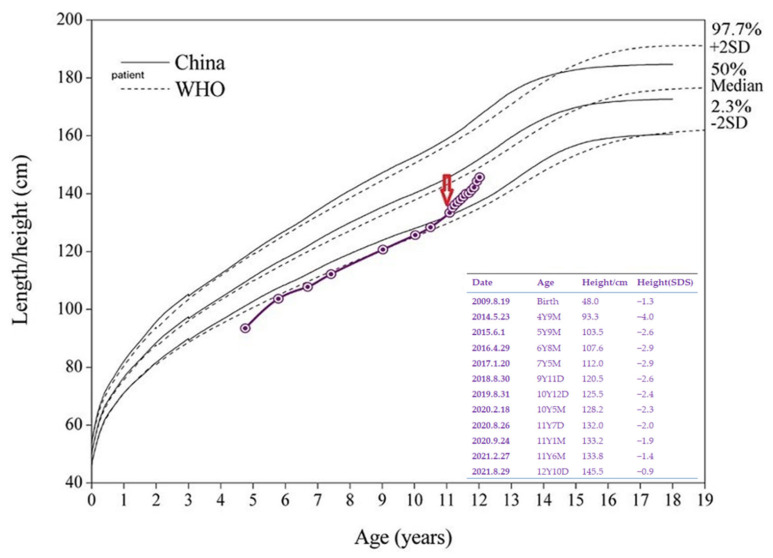
The anthropometric data before and after starting rhGH therapy. The arrows in the growth charts represent the initiation of growth hormone therapy.

## Data Availability

The data supporting the findings of the study are available from the corresponding author upon reasonable request, without undue reservation.

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
