# Peer review of "Recombinant Human Growth Hormone Therapy for Childhood Trichorhinophalangeal Syndrome Type I: A Case Report"

_children, 2022, doi:10.3390/children9101447_

Round 1

Reviewer 1 Report

The use of GH therapy in rare syndromes could be improved because it could be crucial in the patient's life. 

In your study, no real focus emerges and are not highlighted the differences with previous clinical studies that have already investigated the effects of GH therapy in children with pathogenic variants in TRPS1 (such as the recent article of dr. Yael Levy-Shraga, 2020). 

Furthermore, the genetic part needs to be improved (line 115-131) and expanded. Sanger method is not used to verify the pathogenicity of a variant. There is a lack of criteria data according to the ACMG: GNOMAD data, literature data, information from bioinformatics prediction sites (Mutation Taster, SIFT, POliphen) ,  pLOF value, are all missing.

(130-131): It is impossible to establish that the function of the TRPS1 protein disappears even in the presence of nonsense variant without citing a previous study or doing functional studies. 

From an endocrinological point of view, the argument appears discontinuous and not well explained. 

line 192-196: it does not appear that the focus of the work is either to emphasize the importance of genetic investigations in children with suspected TRPS1 (since in reality the use of NGS has been introduced in clinical practice for many years and all international guidelines emphasize the importance of genetic investigations in children with suspected rare syndromes) or to expand the mutational spectrum of the TRPS1 gene (since this case report only reports one new mutation and the work does not go into a study of the functional domains of the protein TRPS1, the variants described so far and their correlation with preferential mutational areas). 

I hope these observations can serve to improve your work. 

Reviewer 2 Report

The authors report an 11 year old child with a rare syndrome and review the scant literature.

The article is well written but I have a few comments.

  1. Write “patient” not, case report.

  2. Mention skeletal age before and after therapy both in text and draw on the growth chart.

  3. When reviewing previously reported patients refer as to whether they had true GHD (ie primed peak after stimulation of below 5µg/ml. If not mentioned it.

The growth of the described patients is similar to ISS patients. Mention this in the discussion and add that there is insufficient proof that acceleration of the growth rate by hGH in non GH deficient children will result in increased adult height.

Is the adult height of the published patients known?

Round 2

Reviewer 1 Report

I think that now your changes are helpful for a better understanding!

-I would change "possible pathogenic nonsense mutation" with "likely pathogenic nonsense variant". 

-in addition, the word "variant" is preferred to the word "mutation", so I suggest changing it in the text.  

I suggest also paying attention to repetitions "mutation-mutation" as in lines 60-61, 63-64, 130-132, 136, but also in a lot of lines in the text. You can use a synonym or change the sentences!

line 60: TRPS I is mainly caused by TRPS1 60 gene mutations

lines 126-127: "Trio whole exome sequencing (Trio-WES) for the proband and his parents using  next generation sequencing (NGS)were performed in our unit".  You can change in "Trio whole exome sequencing (Trio-WES) for the proband and his parents were performed in our unit" 

line 129: The variant was verified (the word "confirmed" is better) by  Sanger sequencing (Fig. 3)

lines 221-223: "We could not study  the effect of the mutation on the expression of TRPS1 because other blood samples and clinical phenotypes in other members of the family were not collected in the later period."

I suggest changing this sentence in this way: "It was not possible to perform functional studies to analyze TRPS1 protein expression." 

It is not useful to collect clinical phenotypes or blood samples of other members of the family because the variant is the novo!
